# Telomere Maintenance Pathways in Lower-Grade Gliomas: Insights from Genetic Subtypes and Telomere Length Dynamics

**DOI:** 10.3390/ijms26094175

**Published:** 2025-04-28

**Authors:** Meline Hakobyan, Hans Binder, Arsen Arakelyan

**Affiliations:** 1Bioinformatics Group, Institute of Molecular Biology National Academy of Sciences of the Republic of Armenia (NAS RA), Yerevan 0014, Armenia; m_hakobyan@mb.sci.am; 2Interdisciplinary Centre for Bioinformatics, University of Leipzig, 04103 Leipzig, Germany; binder@izbi.uni-leipzig.de; 3Armenian Bioinformatics Institute, Yerevan 0014, Armenia

**Keywords:** low-grade glioma, telomere, telomere maintenance mechanisms, telomerase, alternative lengthening of telomeres, pathway signal flow

## Abstract

Telomere maintenance mechanisms (TMMs) play a critical role in cancer biology, particularly in lower-grade gliomas (LGGs), where telomere dynamics and pathway activity remain poorly understood. In this study, we analyzed TCGA-LGG and CGGA datasets, focusing on telomere length variations, pathway activity, and survival data across IDH subtypes. Additional validation was performed using the GEO COPD and GBM datasets, ensuring consistency in data processing and batch effect correction. Our analysis revealed significant differences in TEL pathway activation between Short- and Long-TL groups, emphasizing the central role of TERT in telomere maintenance. In contrast, ALT pathway activation displayed subtype-specific patterns, with IDH-wt tumors exhibiting the highest ALT activity, primarily driven by the RAD51 branch. Validation using CGGA data confirmed these findings, demonstrating consistent TEL and ALT pathway behaviors across datasets. Additionally, genetic subtype analysis revealed substantial telomere length variability associated with ATRX and IDH mutation status. Notably, IDHwt-ATRX WT tumors exhibited the shortest telomere length and the highest ALT pathway activity. These findings highlight distinct telomere regulatory dynamics across genetic subtypes of LGG and provide new insights into potential therapeutic strategies targeting telomere maintenance pathways.

## 1. Introduction 

The traditional classification of central nervous system (CNS) tumors has relied on histological assessments. Historically, tumors classified as World Health Organization (WHO) grades 1 and 2, characterized by slower growth rates, have been referred to as “low-grade gliomas” (LGGs) [1]. In contrast, tumors of WHO grades 3 and 4, exhibiting more aggressive and rapid growth, have been categorized as “high-grade gliomas” [1]. However, recent advancements emphasize the crucial role of molecular biomarkers, offering both supplementary and definitive diagnostic insights [2]. The WHO classification system for CNS tumors has evolved to incorporate key genes and proteins, fundamentally shaping a more comprehensive understanding of these neoplasms [3].

However, in the revised taxonomy of gliomas [3], assessing LGGs poses a challenge. In the intricate landscape of adult-type diffuse gliomas, distinctions arise among IDH-mutant and 1p/19q-non codeleted astrocytoma (IDH-A), IDH-mutant and 1p/19q-codeleted oligodendroglioma (IDH-O), and IDH-wildtype glioblastoma (IDH-wt) [3]. Moreover, astrocytomas are characterized typically by co-mutations in Alpha Thalassemia/Mental Retardation Syndrome X-Linked (*ATRX*) and *TP53* genes, while oligodendrogliomas mostly feature activating mutations in the Telomerase Reverse Transcriptase (TERT) gene promoter.

The nearly mutually exclusive mutations of ATRX in astrocytomas and of TERTp in oligodendrogliomas are striking, especially because both genes are key players in TMMs, where the activation of TERT is central in the canonical telomerase-dependent TMM (TEL) [4], while ATRX deactivation seems to be an important factor for inducing the alternative lengthening of telomeres TMM (ALT) [5]. Hence, IDH-A and IDH-O LGG subtypes seem to differ in their TMMs, with dominant ALT in the former and dominant TEL in the latter. In contrast, in IDH-wt glioblastoma host TERTp, ATRX mutated as well as wild-type tumors regarding these two genes.

The interplay between genetic alterations and telomere maintenance mechanisms has emerged as a focal point for understanding the underlying biology of gliomas. The TMM is employed by cells with unlimited proliferative potential to preserve the length of their telomeres [6,7]. Telomerase activity is rarely observed in normal brain tissues; however, malignant gliomas often show extensive involvement of telomere-related changes [8] via the two recognized telomere maintenance mechanisms, either TEL- and/or ALT-TMM [5].

TERT, a catalytic subunit of telomerase [4], and ATRX [9], an ALT-associated chromatin remodeler [10], are integral components of the intricate machinery governing telomere lengthening and maintenance. Tumors carrying *TERT* promoter mutations demonstrated an increased expression level of *TERT* compared to wild-type tumors, suggesting a correlation between the mutated promoter and the upregulation of TERT in adult gliomas [11]. Furthermore, ALT-positive cells are characterized by heterogeneous telomere lengths, a hallmark of the telomerase-independent ALT pathway [5]. 

The correlation between TERT promoter mutations and increased TERT expression in tumors, heterogeneous telomere lengths in ALT+ cells, and the genetic and molecular diversities across subtypes, including IDH mutations, highlights the complexity of TMMs in LGG. These findings underscore the critical role of the interplay between genetic alterations, telomere length, and TMMs in glioma biology, emphasizing the need for further investigation to enhance diagnostic accuracy and develop targeted therapeutic strategies. Moreover, a study has shown that LGGs have the highest frequency of ATRX mutations (28%) among all cancers [12]. However, an open question remains regarding the distinction between TERT promoter mutations and ATRX mutations, as well as their implications for telomere maintenance mechanisms.

Recently, we studied the activity of the TEL- and ALT-TMM pathways in pan-cancer settings, showing that TEL and ALT pathway activities vary across different cancers, with ALT showing greater variability compared to TEL [13]. This study aims to evaluate alterations in telomere maintenance mechanisms within LGG tumors, with a particular focus on the elucidation of the relations between TMM pathway branches, telomere length, glioma mutation profiles, and subtypes. Given the heterogeneous nature of LGGs, understanding the nuances of telomere regulation in LGGs may provide crucial insights for targeted therapeutic strategies and prognostic assessments.

## 2. Results

### 2.1. Association of Telomere Length, TMM, and IDH Mutation in LGG Samples

Telomere length data, as previously defined by Barthel et al., (2017) [14] were utilized to classify TCGA-LGG samples into two categories: “long telomere length” (Long TL) for tumors where the telomere length exceeded that of the paired blood samples and “short telomere length” (Short TL) for tumors with shorter telomeres compared to normal tissue. The distribution of samples according to telomere length and IDH mutation status is presented in Table 1. 

Building on our previous work, which established a framework for assessing TEL and ALT pathways using TMM signatures and the PSF algorithm [13], we calculated the activity scores of the TEL and ALT pathways to explore telomere regulation dynamics in LGGs.

Comparative analysis of TEL and ALT pathway activation relative to telomere length revealed a significant difference (*p* = 0.034, Mann–Whitney U test) in TEL pathway activation (measured by PSF) between Long-TL and Short-TL groups, with the Short-TL group showing a slightly higher median activation (Figure 1A). In contrast, the ALT pathway did not show a significant difference (*p* = 0.48, Mann–Whitney U test) in activation between long- and short-telomere groups (Figure 1B). Moreover, no significant correlation was observed between the telomere length (TL) ratio and the ALT or TEL pathways (Figure 1C,D).

To evaluate the effect of TMM pathway nodes, we individually analyzed the PSF signal at each node by extracting PSF signals and comparing them between the Short-TL and Long-TL groups (Figure 2A,B; Appendix A). The TEL pathway demonstrates the central role of the TERT complex in telomere maintenance. In the Short-TL group, the PSF activity of the TERT node was significantly higher than in the Long-TL group (Mann–Whitney U test, *p* = 2.9 × 10^−15^; Figure 2A). Moreover, regardless of the telomere length in LGG samples, key interactions of the TEL-TMM pathway involve the activation of TERT and TERT nuclear import nodes (Appendix A). Furthermore, in samples with long telomeres, key interactions involving TERT and TERC, as well as TERT nuclear import nodes, and low levels were noticed in the telomerase branch compared to short telomeres (Figure 2A; Appendix A).

In the ALT pathway, significant differences between the Short- and Long-TL groups were observed in Holiday junction (HJ) Dissolution (Mann–Whitney U test, *p* = 0.0035; Figure 2B). Despite these variations, the overall activity of the ALT pathway, as indicated by the final sink node, remains similar for both long and short telomere lengths.

The analysis of telomere length dynamics and pathway activity across IDH subtypes, with a particular focus on the TEL and ALT pathways, was conducted on 183 samples with a known IDH status. The complete distribution of these samples by IDH subtype and telomere length is presented in Table 1. Under short-telomere conditions, we observed elevated PSF activities in TEL pathway components, particularly TERT and telomerase, in the IDH-O and IDH-A groups. A decrease in activity was noted in the IDH-A subtype, specifically for TERT (Figure 3A; Appendix A). The PSF values of TEL pathway branches were higher in the IDH-A and IDH-O subtypes, with short telomeres across most branches. In contrast, the IDH-wt subtype exhibited marked upregulation of TEL pathway activity compared to the other subtypes (Figure 3A; Appendix A). Although no significant differences were observed between the mentioned groups, these trends highlight the distinct alterations in TEL pathway activities across subtypes.

The ALT pathway showed a heightened response to telomere elongation, with increased involvement of HJ Dissolution and Resolution (Figure 3B). The ALT pathway exhibited activation of most branches under long-telomere conditions in the IDH-A Strand Invasion, HJ Resolution, and HJ Dissolution groups (Figure 3B). Conversely, in the IDH-wt subtype, we observed increased ALT pathway branch activity in both the long and short cohorts (Figure 3B, Appendix A).

The CGGA data results were consistent with those observed in TCGA, where the IDH-A, IDH-O, and IDH-wt subtypes demonstrated similar TEL pathway activation patterns, with TERT being the main contributor to telomerase activity (Appendix A). The ALT pathway in CGGA also mirrored TCGA findings, with IDH-wt tumors showing the highest ALT PSF scores among the IDH-A, IDH-O, and IDH-wt subtypes (Appendix A). In this case, the primary driver of ALT activity was the RAD51 branch, underscoring its crucial role in alternative telomere maintenance in IDH-wt tumors.

### 2.2. Telomere Length Variability Across Genetic Subtypes

Our analysis revealed differences in the telomere length (TL) ratio and ALT pathway PSF scores across various genetic subtypes, particularly concerning ATRX and IDH status. More specifically, a significant decrease in the TL ratio was observed between the IDH-A-ATRX Mutant group and other subtypes, with the most notable reduction in TL ratios seen in IDH-O-ATRX-WT tumors (*p* < 2.22 × 10^−16^, Mann–Whitney U test) and IDHwt- ATRX-WT (*p* = 1.6 × 10^−7^, Mann–Whitney U test) (Figure 4A). Moreover, the IDHwt-ATRX WT and IDHwt-ATRX Mutant groups exhibited relatively low TL ratios, consistent with the absence of active ALT mechanisms in these tumors. IDH-A-ATRX Mutant tumors showed a substantial increase in TL ratio. The ALT activity differs significantly across subtypes. The highest ALT PSF values were observed in the IDHwt-ATRX WT and IDHwt-ATRX Mutant samples, with a notable *p*-value of 6.9 × 10^- 14^ and *p*-value 0.001 (Mann–Whitney U test) compared to the IDH-A-ATRX Mutant samples (Figure 4B). In contrast, the IDH-O-ATRX-WT and IDH-O-ATRX Mutant subgroups exhibited the lowest ALT activities, consistent with the absence of active ALT mechanisms in these tumors (Figure 4B). The IDH-A-ATRX Mutant and IDH-A-ATRX-WT subgroups displayed intermediate ALT activity levels, with statistically significant differences observed between certain groups (*p* = 1.3 × 10^−7^ and *p* = 0.0066, Mann–Whitney U test).

### 2.3. Survival Analysis Based on Telomere Length and TEL/ALT Pathway Activation

Kaplan–Meier survival curves were generated to assess the impact of telomere length and the activity of TEL and ALT pathways on OS and PFS in LGG patients (Figure 5A,B). Patients with long telomeres exhibited significantly different overall survival outcomes based on TMM phenotypes (*p* = 0.0023) (Figure 5A). Specifically, the groups with high ALT and low TEL activities and high ALT and high TEL activities had the poorest OS. In contrast, the ALT-middle/TEL-middle group had the best overall survival (Figure 5A). In patients with short telomeres, the OS also varied significantly across the groups (*p* < 0.0001). The group with ALT-high/TEL-high phenotype showed the worst overall survival, while the group with low ALT and low TEL activities had better overall survival outcomes. PFS significantly differed among the groups with long and short telomeres (Long TL *p* = 0.018; Short TL *p* = 0.012) (Figure 5B). The ALT-high/TEL-high phenotype had shorter PFS for both telomere groups, whereas the ALT-middle/TEL-middle phenotype demonstrated longer PFS in the long telomere group (Figure 5B). Overall, the TMM-related prognoses are in line with the well-documented OS prognoses of the LGG subtypes, which are worst for IDH-wt, at the middle level for IDH-A, and best for IDH-O [15] (Appendix A). Hence, genetic, epigenetic, and transcriptomic factors governing these LGG subtypes systematically affect their TMMs, possibly impacting their prognosis. Multivariate Cox proportional hazards models were applied to evaluate the impact of telomere phenotypes (ALT/TEL status) on both the OS and PSF, adjusting for age, treatment modalities, and telomere length. For OS, both ALT-high/TEL-low (HR = 5.77; 95% CI: [1.97–16.88]; *p* = 0.001) and ALT-high/TEL-high (HR = 5.15; 95% CI: [2.00–13.26]; *p* < 0.001) were significantly associated with poor survival (Appendix A). For PFS, the ALT-high/TEL-low group demonstrated a significantly increased risk of disease progression compared to the reference group (ALT-low/TEL-low), with a hazard ratio (HR) = 5.88, 95% CI: [2.27–15.20], and *p* < 0.001 (Appendix A).

Notably, telomere length alone was not significantly associated with either PFS or OS.

### 2.4. Validation Through Negative Control Analysis

We performed a negative control analysis to validate the reliability of our TMM methodologies. We specifically examined the PSF activity of the TEL and the ALT pathways of the COPD and COPD control (control_COPD) samples compared to GBM, GBM normal samples (N_GBM) from TCGA-GBM, and LGG samples. Our findings demonstrate statistically significant differences in both TEL and ALT values among the COPD, COPD control groups versus GBM, and LGG samples (TEL *p* < 0.0001, Mann–Whitney U test; ALT *p* < 0.05, Mann–Whitney U test, for all comparisons) (Figure 6A,B). These results serve to highlight the inherent variability of telomere dynamics in different conditions, ensuring that our subsequent TMM analyses can accurately discern true biological changes from background variation.

## 3. Discussion

While telomere length and TMMs have been studied extensively in the context of cancer, the relationship between telomere length and glioma risk remains unclear [16,17]. Our study provides comprehensive insights into the dynamics of telomere length and TMMs within LGGs, with a focus on the differences between various IDH subtypes and their clinical implications.

### 3.1. Telomere Length and TMM Pathway Dynamics

Our results demonstrate that alterations in telomere length are associated with differential activation of the TEL pathways in LGGs. Specifically, tumors with short telomeres showed higher TEL pathway activation than those with long telomeres, while ALT pathway activation did not significantly differ between the telomere length groups. This suggests that TEL pathway activation may be more responsive to shortening in telomere length, potentially reflecting the telomere maintenance mechanisms that are triggered under critically short-telomere conditions [18]. This result aligns with the established role of the TEL pathway in maintaining telomere integrity through telomerase activity, especially in telomeres that have undergone significant shortening [19]. The study of 21 thyroid tumors found that increased telomerase activity and reduced telomere length are significantly associated with the development of thyroid tumors [20]. In contrast, the ALT pathway did not show a statistically significant difference in activation between long- and short-telomere groups. This finding suggests that ALT activation may not be as closely linked to the immediate telomere length status, possibly due to the distinct mechanisms through which ALT maintains telomeres, such as recombination-based processes that may be independent of telomere length at a given time point. Studies have shown that ALT-positive cells often exhibit heterogeneous telomere lengths, further indicating that the pathway can operate independently of telomere shortening [21,22]. The absence of a significant difference in ALT activation also indicates that ALT and TEL pathways might operate through separate regulatory mechanisms. ALT activation may not be driven solely by telomere shortening but could be influenced by other cellular stressors or genomic instabilities. 

Moreover, the TEL pathway demonstrates higher PSF values for key TERT interactions and nuclear import in long telomeres, while short telomeres show elevated PSF values for the TERT complex. Several studies have established that TERT is a key regulator of TEL-TMM, as its activation is essential for maintaining telomere length in most cancers through telomerase activity [4,23]. In contrast, the ALT pathway reveals enhanced recruitment to Holiday junction Dissolution in short telomeres, with overall ALT activity being slightly higher in long telomeres, highlighting distinct maintenance dynamics based on telomere length (Appendix A).

This raises an intriguing question regarding the causal relationship between ALT and telomere length. Specifically, does the activation of ALT result in the elongation of telomeres, or is telomere lengthening a consequence of ALT activity?

### 3.2. Subtype-Specific Variations in TEL and ALT Pathway Activation

IDH mutations are commonly found in human cancers. In gliomas, they are present in over 80% of grade II cases (LGG) and grade IV (GBM), up to 73% of cases of secondary GBM, and 3.7% of primary GBM cases [24,25,26]. Consequently, the telomere length variation has been associated with the IDH status [27]. However, our results show that there is no clear relation between IDH status, telomere length, and TMM pathway activation.

The significant prevalence of IDH mutations in these tumors highlights their critical role in the understanding of IDH-subtype-specific telomere length variations. Under conditions of short telomere length, key components of the TEL pathway, particularly TERT and telomerase synthesis, were strongly activated in the IDH-A subtypes, suggesting a predominant reliance on telomerase-mediated telomere extension. However, as telomeres lengthened, there was a marked reduction in TEL pathway branch component activity, especially in these subtypes, reflecting the sensitivity of telomerase-based mechanisms to telomere attrition. In the IDH-wt subtype, short-telomere conditions showed elevated activity in most branches. In contrast, the ALT pathway showed a robust response to long telomere conditions, particularly through increased activation of HJ Dissolution and Resolution in the IDH-A subtype. On the other hand, the IDH-wt subtype demonstrated moderate TEL pathway activity and the highest ALT pathway activation among all subtypes in both datasets analyzed, suggesting a greater reliance on ALT mechanisms. One plausible explanation for the observed high ALT activity in IDH-wt tumors could be related to the incompleteness or inaccuracies of the ALT pathway․ Alternatively, the heightened ALT activity observed in IDH-wt tumors might be attributed to the inherent heterogeneity in ALT pathway activation and telomere length dynamics, with a more prominent role for the RAD51 branch than previously thought, as all significant results point to this branch as the main driver of ALT activity in our analyzed samples. This underscores the need for further evaluation of RAD51’s role in ALT pathway activation, opening up new perspectives for understanding its contributions.

Furthermore, our previous study indicated that ATRX-mutated samples exhibited lower ALT PSF values across several cancer types, including PAAD, KIRC, and LGG. In contrast, we observed that ATRX-mutated samples in BRCA, UCEC, and COAD displayed elevated ALT pathway activity [28]. Specifically, while some IDH-mutant tumors displayed higher ALT values compared to IDH-wt tumors, this trend was not universally observed across all samples (Appendix A). In particular, COAD, PAAD, and LGG (Appendix A) showed elevated ALT PSF values in IDH-wt tumors, highlighting the complexity of ALT pathway activity in different tumor contexts. This highlights the non-definitive role of ATRX mutations in determining ALT pathway activation, as the mutation does not universally predict heightened ALT activity across various cancer types. Moreover, there is an ongoing debate regarding the role of ATRX in ALT activation [29,30]. An integrative analysis of 2658 tumors across 38 primary sites revealed that ATRX/DAXX mutations, although linked to TMMs, are infrequent and inconsistently present in ALT-positive tumors, indicating that ALT activation is not solely driven by these mutations [29]. Furthermore, experimental ATRX knockout in four telomerase-positive, ALT-negative glioma lines showed that only two lines exhibited ALT features post-ATRX loss, highlighting the need for additional factors beyond ATRX in fully establishing ALT activity [31].

Notably, tissue-specific discrepancies in ATRX–ALT associations, such as the contrasting trends observed in BRCA, KIRC, and UCEC versus LGG, indicate a more complex and context-dependent regulatory environment. These differences may be shaped by epigenetic states, cell-of-origin influences, and the co-mutational environment. BRCA and UCEC, both characterized by frequent TP53 mutations and hormonal signaling pathways, may present permissive conditions for ALT activation despite ATRX disruption [32]. Furthermore, the tumor microenvironment may also play a significant role in modulating ALT pathway dynamics. Hypoxia has been shown to induce DNA damage and replication stress, conditions that favor ALT-associated phenotypes [33].

Likewise, immune-related signals and inflammatory cytokines may indirectly influence telomere maintenance by affecting genomic stability and the DNA repair pathway [34]. Differences in stromal composition and immune cell infiltration across tissue types may, thus, contribute to the heterogeneous activation of ALT despite similar genetic alterations. Together, these findings underscore the importance of integrating tissue-specific factors, co-mutation patterns, and microenvironmental influences when interpreting the functional consequences of ATRX alterations in cancer.

### 3.3. Telomere Length Variability Across Genetic Subtypes

In order to better understand our results observed in IDH-wt tumors with high ALT activity, we conducted more detailed analyses, highlighting the nonlinear relationship between ATRX status and high ALT activity.

The most pronounced TL ratio alterations were observed in tumors harboring ATRX mutations, irrespective of IDH mutation status. These tumors exhibited significantly elevated TL ratios compared to other genetic subtypes, suggesting a critical role for ATRX mutations in telomere length regulation and maintenance in gliomas. In contrast, ALT pathway activity was significantly elevated in IDH-wt ATRX-wt samples, consistent with our ALT pathway analysis, where IDH-wt samples exhibited the highest ALT scores. Despite analyses of four samples identifying ATRX mutations as a significant molecular factor in lower-grade gliomas [35], ALT activation has been associated with ATRX deficiency in only 14–35% of high-grade pediatric gliomas [36]. Furthermore, the ALT pathway is known to exhibit a more intricate structure with multiple biomarkers, and various factors can contribute to elevated ALT activity [21]. This complexity might elucidate the high ALT levels observed in the IDH-wt, ATRX-wt subtype. Factors beyond ATRX mutations may be influencing ALT pathway activation in these samples. In a recent study, it was demonstrated that the ALT mechanism was associated with immunogenic IDH-A phenotypes of varying levels of methylation, while TEL was found in highly proliferative IDH-O and IDH-wt gliomas [37]. Telomere maintenance via TEL mechanisms is typically found in highly proliferating tumors of epithelial phenotypes, such as high-grade melanomas and colon cancer, while ALT is more prone to mesenchymal immunogenic tumors, such as sarcomas undergoing epigenetic reprogramming [18,38,39].

### 3.4. Clinical Implications of Telomere Dynamics in LGG

The survival analysis showed that patients with long telomeres and low ALT pathway activation had better overall survival outcomes, while those with ALT-high/TEL-high and ALT-high/TEL-low phenotypes had poorer prognoses. ALT-high samples were associated with poorer clinical outcomes, consistent with previous findings. In a study of 412 glioma patients categorized into four TMM groups; telomerase, ALT, negative, and ALT+TEL, it was reported that patients in the telomerase group were significantly older compared to those in the other TMM groups [40]. This highlights the potential of integrating telomere length and TMM pathway activity into clinical risk stratification models to predict patient outcomes better and guide treatment decisions. In addition, multivariate Cox proportional hazards models revealed that the ALT-high/TEL-low and ALT-high/TEL-high phenotypes were significantly associated with poorer overall survival and progression-free survival, further emphasizing the prognostic value of TMM pathway activity.

### 3.5. Translational Relevance of TMM-Targeted Therapies in Gliomas

Telomerase inhibitors and other TMM-targeted strategies have been the subject of increasing interest in clinical trials, particularly in cancers where telomere maintenance plays a critical role in tumor progression and therapy resistance. Telomerase inhibitors, such as Imetelstat, have shown promise in early-phase clinical trials, demonstrating potential in cancers with high telomerase activity, including hematological malignancies and certain solid tumors [41]. However, the application of these therapies in cancers like gliomas, where ALT may predominate, presents significant challenges. ALT is a more complex and less well-understood pathway compared to telomerase, and targeting it could require different strategies, such as inhibition of the DNA repair machinery that facilitates ALT, including proteins like *RAD51*, as shown in our research. *RAD51* is frequently overexpressed in various malignant solid tumors and is associated with unfavorable clinical outcomes. It plays a pivotal role in tumor metabolism, metastasis, and resistance to therapy across multiple cancer types. The structure and expression profile of *RAD51*, along with the main regulators, facilitate its function in homologous recombination [42]. In our study, we observed elevated RAD51 branch activity, specifically in IDH-wt tumors with active ALT. This finding suggests enhanced HR activity in these tumors, potentially supporting telomere elongation through ALT-associated recombination mechanisms. Interestingly, this aligns with clinical observations that IDH1-wt tumors often exhibit reduced sensitivity to temozolomide, as previously reported in studies showing a higher rate of temozolomide response in IDH1-mutant low-grade gliomas compared to their IDH1-wt counterparts [43]. The increased RAD51 activity in IDH-wt tumors may contribute to more effective DNA damage repair via HR, thereby conferring resistance to temozolomide-induced cytotoxicity. These findings underscore the need for further mechanistic investigation into *RAD51*’s dual role in ALT activation and DNA repair, which may represent a potential therapeutic target in IDH-wt gliomas.

### 3.6. Validation and Limitations

Our findings are supported by a negative control analysis comparing PSF activity in COPD and COPD control samples to GBM and LGG. The statistically significant differences observed across these groups confirmed the specificity of our TMM methodologies and underscored the variability of telomere dynamics in different biological contexts. However, this study has some limitations, including solely bioinformatics data analysis and the potential for sample selection bias. We also acknowledge that certain subgroups, particularly IDH-wt, contained relatively few samples. This may limit the statistical power of subgroup analyses and affect the generalizability of our findings. Furthermore, the limitations of our study are reflective of broader challenges in the biological field, including data incompleteness, inconsistencies across studies involving different LGG samples, and a general lack of sufficient and appropriate data for comprehensive analyses. Moreover, the IDH subtype information was not available for all LGG samples, which may have contributed to limitations in data generalization. Additionally, the ambiguous role of ATRX in ALT telomere maintenance and the ambiguous interpretation of data pose significant challenges. Moreover, although we used telomere length estimates derived from Illumina whole-genome sequencing data, it is important to acknowledge that NGS-based approaches have inherent limitations in accurately quantifying highly repetitive regions such as telomeres. While tools like TelSeq implement correction strategies (e.g., GC-bias normalization) to improve reliability, more direct experimental methods (e.g., Southern blot and qPCR) generally offer higher precision for telomere length measurement. While the PSF algorithm provides a valuable tool for assessing pathway activity, further validation in independent cohorts and experimental settings is required to confirm these findings.

## 4. Materials and Methods

### 4.1. Data Sources and Preprocessing

In this study, we employed a methodology previously described in our earlier work [13], which involves data preprocessing, the utilization of TMM signatures [44], and the application of the Pathway Signal Flow (PSF) algorithm [45,46].

We specifically focused on the TCGA-LGG dataset, which contains 532 samples in total. However, our analysis was limited to 506 samples for which telomere length data were available. The normalized RNA sequencing counts were extracted from our previous study [13]. The LGG subtype information, based on expression data, was derived from the study by Willscher et al., 2021 [37]. The IDH mutation status and genetic subtypes for TCGA-LGG project samples were obtained from the study by Ceccarelli et al., 2016 [47]. For the analysis of telomere length variations and pathway activity across different IDH subtypes, we used only samples for which the expression group and IDH type assignment were consistent for our analyses in both studies. Samples with ambiguous, or conflicting subtype annotations were excluded to reduce potential classification bias. In total, 182 samples were taken. The distribution of IDH status and telomere length across these samples, including the *p*-values from Fisher’s exact test for IDH groups and the t-test for telomere length, is provided in Table 1. The mean and standard deviation (MTL SD) of the telomere length ratio for each experimental group are also summarized in the table. Telomere length data were obtained from the previous publication by Barthel et al., 2017 [14]. In samples where the length of the tumor telomere surpassed the paired reference telomere length estimated in the blood of the same patient, it was denoted as “long telomere length” (Long TL, TL ratio > 1); otherwise, it was designated as “short telomere length” (Short TL, TL ratio < 1).

Overall (OS) and progression-free (PFS) survival data for TCGA-LGG samples were obtained from cBioportal [48]. TMM phenotype information was derived from our previous work [13]. Specifically, samples with ALT_log PSF > 3.64 were classified as ALT-high, while those with ALT_log PSF < 1.51 were classified as ALT-low. Similarly, samples with TEL PSF > 0.927 were considered TEL-high, and those with TEL PSF < 0.438 were categorized as TEL-low. The thresholds for ALT and TEL were determined using segmented regression, which identified breakpoints in the data. Samples not meeting these criteria were assigned to the intermediate group (ALT-middle/TEL-middle). Based on these definitions, five TMM phenotype groups were constructed for survival analysis: ALT-high/TEL-low, ALT-low/TEL-low, ALT-high/TEL-high, ALT-low/TEL-high, and ALT-middle/TEL-middle.

Ethics approval and consent were not required for this study, as it was performed using publicly available data.

### 4.2. Datasets for Supporting and Validation Analyses

As a validation set for TCGA-LGG findings, we analyzed 176 LGG samples from the Chinese Glioma Genome Atlas (CGGA) [49,50]. These samples were categorized based on their IDH mutation status and chromosomal alterations.

We assigned samples to the IDH-A group if they were classified as astrocytomas, contained a mutation in the IDH gene, and lacked chromosome 1p19q co-deletion. Conversely, the IDH-O group included all IDH-mutant oligodendrocytoma samples with chromosome 1p19q co-deletion. Samples without IDH mutations were classified as IDH-wt, irrespective of their histological type. RNA sequencing data were processed similarly to the TCGA-LGG data to ensure consistency [13]. Briefly, read counts were normalized to the sequencing library size using the DESeq2 R package (version 1.34.0) (https://bioconductor.org/packages/DESeq2/ accessed on 25 February 2025) [51] and subsequently transformed into log values. For the evaluation and correction of batch effects, the SVA R package (version 3.42.0) (https://bioconductor.org/packages/sva/ accessed on 25 February 2025) [52] was employed, and long-transformed data were then mean-centered across all samples and converted to fold-change (FC) values. The calculation of PSF scores was performed using the approach previously described in our publications [13,44,46].

Gene expression data for Chronic Obstructive Pulmonary Disease (COPD) were obtained from the Gene Expression Omnibus (GEO) database under accession number GSE124180. The RNA-seq data were collected from patients with 21 COPD cases and 21 controls [53]. This dataset was used as a “negative” control to compare and analyze the TMM pathway activity in non-cancer and cancer tissues.

COPD, LGG, and glioblastoma (GBM) samples’ raw counts were normalized to the sequencing library size using the DESeq2 (version 1.34.0) [51] R package and converted to log values. The Limma R package (version 3.50.3) [54] was used to evaluate batch effects and their correction. Log-transformed data were mean-centralized over the corresponding dataset and converted to fold-change (FC) values. We used the same set of statistical tests for these analyses as well.

### 4.3. Statistical Analyses

The TEL and ALT pathway branch-level PSF values were extracted from the TMM pathways’ topologies [18,44]. The significance of PSF differences in groups stratified according to telomere length and IDH status was assessed using the Mann–Whitney U test. The Pearson correlation coefficient was used to evaluate the correlation between TL ratio and TEL or ALT PSF values. We used Kaplan–Meier survival curves from the survival (version 3.3.1) (https://cran.r-project.org/web/packages/survival/index.html accessed on 25 February 2025) and survminer (version 0.4.9) (https://cran.r-project.org/web/packages/survminer/index.html accessed on 25 February 2025) R packages to analyze OS and PFS in groups stratified by telomere length, IDH mutation status, and TMM phenotypes. The significance of survival differences was assessed using the log-rank test. *p*-values < 0.05 were considered significant for all statistical tests. Multivariate Cox proportional hazards regression models [55] were employed to estimate hazard ratios in the survival analysis, incorporating variables such as treatment modality, patient age, telomere length, and telomere phenotypes. Age was categorized into two groups—young and old—based on the median age of the cohort (40 years). Treatment modalities were consolidated into broader, clinically interpretable categories, including radiotherapy, hormone therapy, surgery, chemotherapy, targeted therapy, and immunotherapy.

## 5. Conclusions

Our study provides comprehensive insights into telomere length dynamics and TMM in LGG, highlighting their association with IDH mutation status and survival outcomes. We observed that a shorter telomere length is associated with increased activation of the TEL pathway, particularly through TERT, while the ALT pathway exhibited significant variation in specific nodes but remained stable overall across telomere length categories. Notably, IDH-wt tumors displayed the highest ALT pathway activation, primarily driven by RAD51. Telomere length variability was significantly influenced by ATRX and IDH status, with the IDHwt-ATRX WT and IDHwt-ATRX Mutant groups exhibiting the lowest TL ratios and the highest ALT activities. Survival analysis revealed that LGG patients with high ALT and high TEL pathway activities had the worst prognosis, while those with moderate TMM activity exhibited the most favorable outcomes.

## Figures and Tables

**Figure 1 ijms-26-04175-f001:**
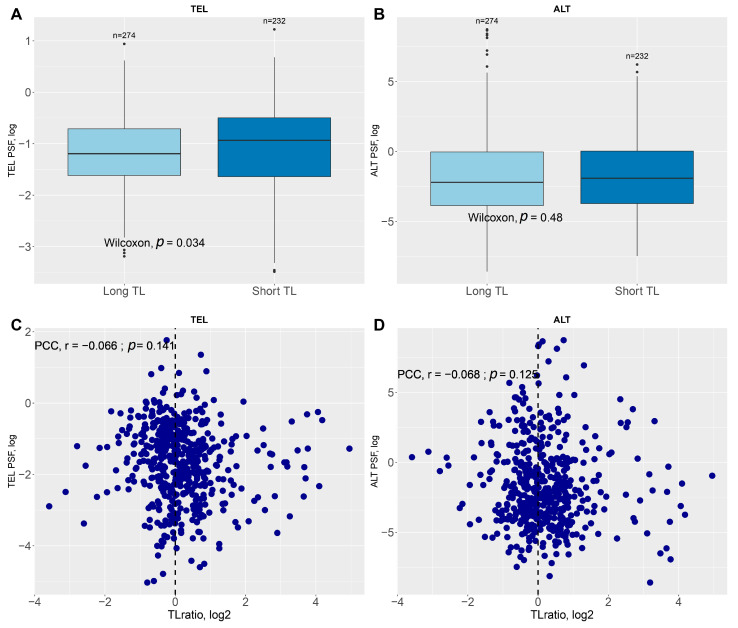
Relationship between telomere length and PSF scores for TEL and ALT pathways. Comparison of (**A**) the TEL PSF and (**B**) the ALT PSF scores between samples with long and short telomere lengths. Statistical significance was evaluated using the Mann–Whitney U test. Scatter plots showing the correlation between the telomere length ratio (TL ratio, log2) and (**C**) the TEL PSF and (**D**) ALT PSF scores. Pearson correlation coefficients (PCCs) and *p*-values are displayed for each analysis.

**Figure 2 ijms-26-04175-f002:**
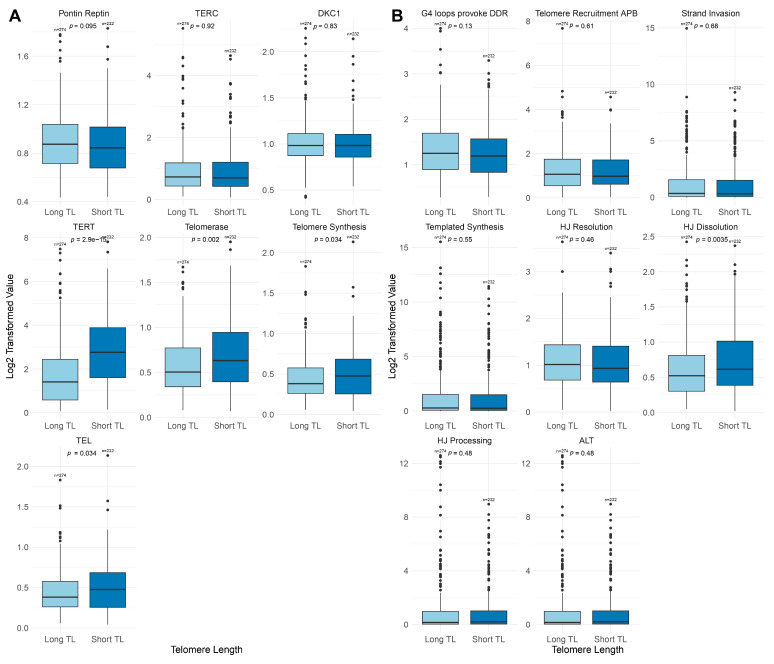
Comparison of telomere lengths between ALT and TEL pathway branches. (**A**) The TEL pathway branches (PSF Log2 transformed), comparing samples with long and short telomere lengths. (**B**) The ALT (PSF Log2 transformed) pathway-associated branches, comparing Long and Short TLs. Statistical significance was assessed using the Mann–Whitney U test. The *p*-values shown indicate the significance of differences between the long- and short-telomere-length groups.

**Figure 3 ijms-26-04175-f003:**
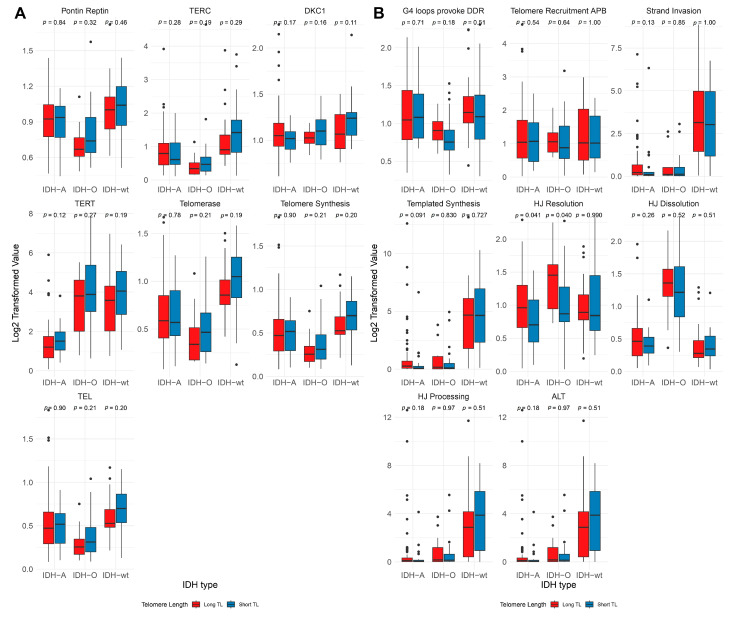
TEL and ALT pathway branch activities across IDH subtypes and telomere length categories. (**A**) The TEL and (**B**) ALT pathway Log2-transformed PSF values of key branches across the IDH subtypes (IDH-A, IDH-O, and IDH-wt). Data are stratified based on telomere length groups (Long TL in red and Short TL in blue). Statistical comparisons were performed between telomere length groups within each IDH subtype. Statistical significance was assessed using the Mann–Whitney U test.

**Figure 4 ijms-26-04175-f004:**
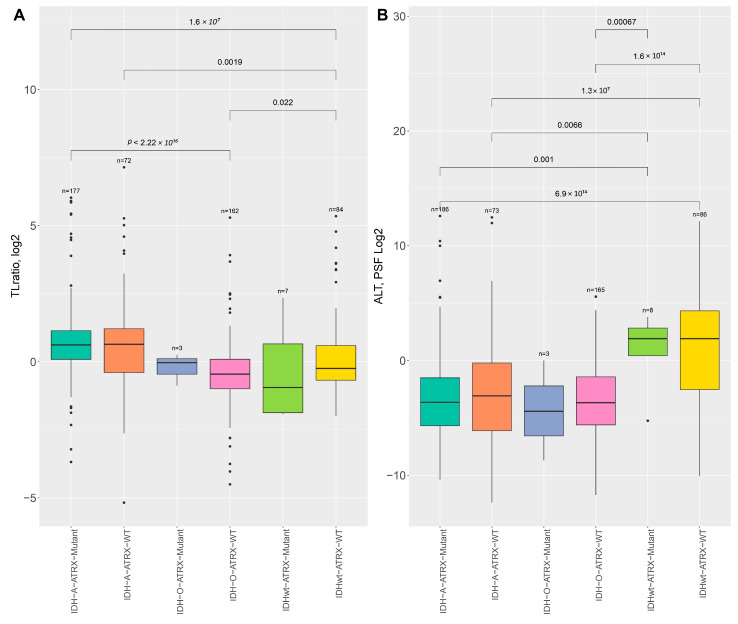
Comparison of TL ratios and ALT pathway PSF scores across IDH genetic subtypes and ATRX status. (**A**) Comparison of TL ratio (log2) across different genetic subtypes, categorized by the IDH and ATRX mutation status. (**B**) Comparison of ALT pathway PSF scores (log2) across IDH genetic subtypes and ATRX status. Statistical significance was evaluated using the Mann–Whitney U test.

**Figure 5 ijms-26-04175-f005:**
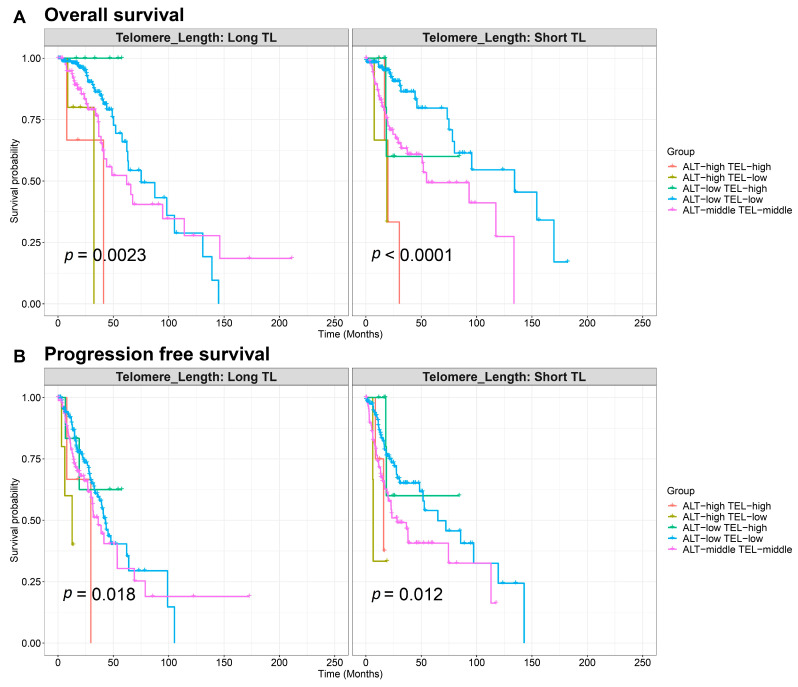
Survival plots for long- and short-telomere groups for TMM phenotype. (**A**) Overall survival and (**B**) progression-free survival curves. The pairwise log-rank test was used to assess the significance between TMM phenotype groups: ALT-high (ALT_log PSF > 3.64), ALT-low (ALT_log PSF < 1.51), TEL-high (TEL PSF > 0.927), and TEL-low (TEL PSF < 0.438). Samples not meeting these criteria were assigned to the intermediate group, ALT-middle/TEL-middle.

**Figure 6 ijms-26-04175-f006:**
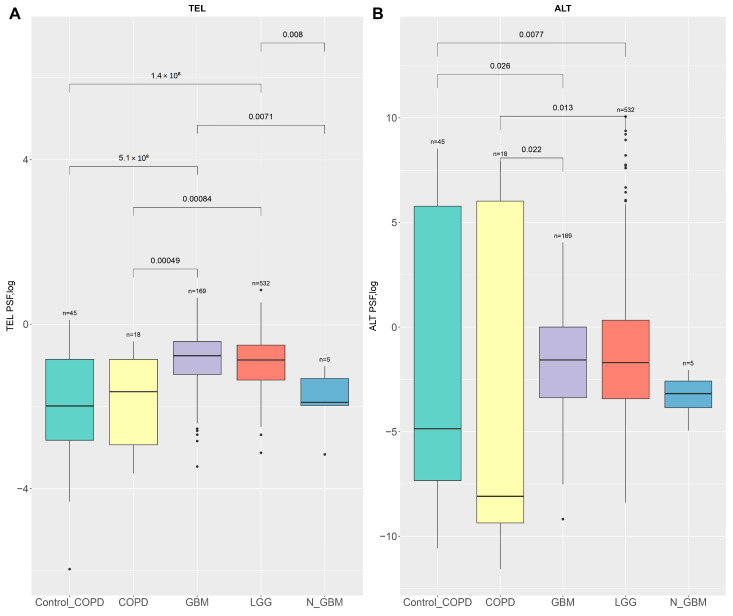
TEL and ALT pathway validation using negative control. (**A**) The TEL pathway and (**B**) ALT pathway PSF scores (log2) across different groups: Control_COPD, COPD, GBM, LGG, and Normal GBM (N_GBM). Notably, in ALT and TEL pathways, activity is significantly higher in GBM and LGG samples compared to Control_COPD and COPD groups.

**Table 1 ijms-26-04175-t001:** Distribution of LGG samples across IDH groups based on telomere length.

	Long TL	Short TL	Long TL Fraction	*p* (Fisher’s Exact Test)	MTL (Mean +/− SD)	*p* (*t*-Test)
LGG	274	232	0.54			
EXP Group *						
IDH-A	73	23	0.76	6.00 × 10⁻^7^	0.55 ± 1.01	0.05683
IDH-O	12	33	0.27	6.19 × 10⁻^7^	−0.41 ± 1.06	0.3284
IDH-wt	22	19	0.54	0.4748	0.19 ± 1.05	0.7108

* Note: only samples with consistent expression groups and IDH-type assignments were included in the analysis.

## Data Availability

The data generated in this study are available in the article and its Appendix A. The data and materials used in this study are available upon request from the corresponding authors. The data generated in this study are publicly available in the Zenodo open repository (https://doi.org/10.5281/zenodo.14730657 accessed on 24 February 2025).

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
