# Peer review of "Telomere Maintenance Pathways in Lower-Grade Gliomas: Insights from Genetic Subtypes and Telomere Length Dynamics"

_ijms, 2025, doi:10.3390/ijms26094175_

Round 1
Reviewer 1 Report
Comments and Suggestions for Authors
1. We recommend clarifying the inclusion/exclusion criteria and clinical feature matching for the TCGA-LGG dataset to address potential selection bias. Additionally, we suggest discussing how the relatively small sample size in subgroups (e.g., IDH-wt) might influence statistical power and generalizability of the findings.
2. We encourage the authors to incorporate multivariate Cox proportional hazards models in the survival analysis to adjust for confounders (e.g., age, treatment history). It would be helpful to explicitly define the thresholds used for categorizing "ALT-high" or "TEL-high" phenotypes in Kaplan-Meier analyses to enhance reproducibility.
3. We recommend expanding the discussion on the tissue-specific discrepancies in ATRX-ALT correlations (e.g., BRCA/UCEC vs. LGG). Exploring potential mechanisms, such as microenvironmental interactions or co-mutational landscapes, would strengthen the interpretation of these findings.
4. We encourage the authors to delve deeper into the mechanistic role of the RAD51 branch in ALT activation within IDH-wt tumors. Integrating literature on RAD51’s involvement in DNA repair and telomere recombination would provide valuable context for its therapeutic potential.
5. We recommend discussing the current state of TMM-targeted therapies (e.g., telomerase inhibitors in clinical trials) to contextualize the translational relevance of the findings. Highlighting challenges in applying these strategies to LGG would add practical depth to the conclusions.
Comments on the Quality of English LanguageThe English is fine and could be improved to more clearly express the research.
Author Response
Reviewer 1 Comment 1: We recommend clarifying the inclusion/exclusion criteria and clinical feature matching for the TCGA-LGG dataset to address potential selection bias. Additionally, we suggest discussing how the relatively small sample size in subgroups (e.g., IDH-wt) might influence statistical power and generalizability of the findings.
Reviewer 1 Response 1: We thank the reviewer for this insightful comment. Regarding the inclusion and exclusion criteria, we used TCGA-LGG samples for which telomere length estimation and complete molecular annotation (including IDH mutation, 1p/19q co-deletion, and ATRX status) were available. Samples with missing or ambiguous annotations were excluded to ensure consistency across analyses. Additionally, samples with ambiguous or conflicting subtype annotations were excluded to reduce potential classification bias, as noted in the revised Materials and Methods – Data Sources and Preprocessing section [Pg 14, Ln 45-47 ].
We also acknowledge the relatively small sample sizes in certain subgroups, particularly the IDH-wt group. This may limit the statistical power of subgroup analyses and affect the generalizability of our findings. We have now explicitly included this point in the Discussion – Limitations section to address the concern raised. [Pg 14, Ln 14-17 ].
Reviewer 1 Comment 2. We encourage the authors to incorporate multivariate Cox proportional hazards models in the survival analysis to adjust for confounders (e.g., age, treatment history). It would be helpful to explicitly define the thresholds used for categorizing "ALT-high" or "TEL-high" phenotypes in Kaplan-Meier analyses to enhance reproducibility.
Reviewer 1 Response 2: We appreciate the reviewer’s insightful comment. In response, we have now incorporated multivariate Cox proportional hazards models into the survival analysis as detailed in the Results Survival Analysis Based on Telomere Length and TEL/ALT Pathway Activation section [Pg 8, Ln 20-21 and Pg 9 1-8], Discussion Clinical Implications of Telomere Dynamics in LGG section [Pg 13, Ln 27-31], Materials and Methods Statistical Analyses section [Pg 16, Ln 7-13], and Supplementary figure S7.
We appreciate the reviewer’s suggestion to clarify the definition of "ALT-high" and "TEL-high" phenotypes used in our Kaplan-Meier analyses. We have now included explicit threshold values for ALT and TEL scores in the revised manuscript Materials and Methods – Data Sources and Preprocessing section and Figure 5 legends [Pg 15, Ln 8-16].
Reviewer 1 Comment 3: We recommend expanding the discussion on the tissue-specific discrepancies in ATRX-ALT correlations (e.g., BRCA/UCEC vs. LGG). Exploring potential mechanisms, such as microenvironmental interactions or co-mutational landscapes, would strengthen the interpretation of these findings.
Reviewer 1 Response 3: We appreciate the reviewer’s suggestion regarding the need to expand on tissue-specific discrepancies in ATRX–ALT correlations. In response, we have revised the discussion section to include a more detailed explanation of the potential mechanisms underlying these differences. The revised text can be found in the Discussion section [Pg 12, Ln 29-44].
Reviewer 1 Comment 4: We encourage the authors to delve deeper into the mechanistic role of the RAD51 branch in ALT activation within IDH-wt tumors. Integrating literature on RAD51’s involvement in DNA repair and telomere recombination would provide valuable context for its therapeutic potential.
Reviewer 1 Response 4: We appreciate the reviewer’s suggestion. In response, we have expanded the Discussion section "Translational Relevance of TMM-Targeted Therapies in Gliomas" of the manuscript to further elaborate on the mechanistic role of the RAD51 branch in ALT activation within IDH-wt tumors.[Pg 13, Ln 32-51 and Pg, 14 1-6 ].
Reviewer 1 Comment 5: We recommend discussing the current state of TMM-targeted therapies (e.g., telomerase inhibitors in clinical trials) to contextualize the translational relevance of the findings. Highlighting challenges in applying these strategies to LGG would add practical depth to the conclusions.
Reviewer 1 Response 5: We thank the reviewer for this valuable suggestion. In response, we have expanded the Discussion section to include a new subsection titled "Translational Relevance of TMM-Targeted Therapies in Gliomas." [Pg 13, Ln 32-51 and Pg, 14 1-6 ].
Reviewer 2 Report
Comments and Suggestions for Authors
In the paper entitled “Telomere Maintenance Pathways in Lower-Grade Gliomas: Insights from Genetic Subtypes and Telomere Length Dynamics”, the authors attempt to correlate the genetic background of specific low grade gliomas subtypes, with telomere maintenance mechanisms and telomere’s length. The authors start by dividing their cohort of samples into long and short telomeres based on a previous analysis published by Barhtel and coauthors in 2017, which analyze sequencing data obtained by illumina NGS. One of the big problems in sequencing telomeric repeats is the fact that illumina NGS sequencing relies on reads alignment, which severely affects the proper quantification of repeated sequences. It is now clear to all the scientific community working on telomeres that the NGS seq data is not enough accurate for telomere length estimation. Furthermore, while TERT promoter mutations associated with telomerase reactivation are now well consolidated, it is still debated if genetic mutations in genes associated with ALT TMM (ATRX/DAXX/p53/H3.3) are sufficient to prove ALT activity. Taking into account these two major considerations, in my opinion the conclusions presented by these authors base on unreliable data and could be misleading for the readers.
Author Response
Reviewer 2 Comment 1:The authors start by dividing their cohort of samples into long and short telomeres based on a previous analysis published by Barhtel and coauthors in 2017, which analyze sequencing data obtained by illumina NGS. One of the big problems in sequencing telomeric repeats is the fact that illumina NGS sequencing relies on reads alignment, which severely affects the proper quantification of repeated sequences. It is now clear to all the scientific community working on telomeres that the NGS seq data is not enough accurate for telomere length estimation.
Reviewer 2 Response 1:
We thank the reviewer for raising this important point regarding the accuracy of telomere length (TL) estimation from sequencing data. While we agree that short-read Illumina-based NGS platforms pose certain challenges in quantifying telomeric repeats, we would like to clarify that our study relies on high-confidence whole-genome sequencing (WGS)-based TL measurements, as described in the source paper by Barthel et al. (2017). Specifically, TL calling was performed using processed BAM files through a dedicated telomere length estimation pipeline, and only samples meeting stringent quality criteria (e.g., >20 million reads) were retained for analysis. Furthermore, the core dataset (n=2,018) used linear mixed modeling to adjust TL for potential confounding effects across tumor and normal tissues, including data from multiple tissue types. An extended dataset (n=6,835) was also employed to increase statistical power, incorporating TL estimates from overlapping low-pass and exome sequencing data, which were carefully integrated and controlled for bias. Moreover, TL quantification was performed using the TelSeq tool, which has been specifically developed to address such limitations. TelSeq identifies telomeric reads based on the presence of a minimum number of TTAGGG repeats and estimates TL by normalizing the abundance of these telomeric reads against a size factor derived from reads with GC content between 48–52%—a range that closely reflects the GC composition of telomeric regions. This correction significantly mitigates amplification biases introduced during library preparation and sequencing. Importantly, the TelSeq method has been validated in the original study (Ding et al., 2014) [1] by demonstrating a strong correlation with gold-standard Southern blot mTRF measurements across 260 leukocyte samples from the TwinsUK cohort.
We have now explicitly included this point in the Discussion – Limitations section to address the concern raised by the reviewer regarding the accuracy of telomere length estimation from Illumina sequencing data. [Pg 14, Ln 23-28].
1. Estimating Telomere Length from Whole Genome Sequence Data | Nucleic Acids Research | Oxford Academic Available online: https://academic.oup.com/nar/article/42/9/e75/1249448 (accessed on 16 April 2025).
Reviewer 2 Comment 2: Furthermore, while TERT promoter mutations associated with telomerase reactivation are now well consolidated, it is still debated if genetic mutations in genes associated with ALT TMM (ATRX/DAXX/p53/H3.3) are sufficient to prove ALT activity.
Reviewer 2 Response 2: We thank the reviewer for highlighting the ongoing debate regarding the sufficiency of ATRX/DAXX/p53/H3.3 mutations in conclusively determining ALT pathway activity. We fully agree that these mutations alone are not definitive indicators of ALT activation, and we have explicitly addressed this complexity in the Discussion section of our manuscript.
Specifically, we state that the Discussion section [Pg 12, Ln 19-21] “this highlights the non-definitive role of ATRX mutations in determining ALT pathway activation, as the mutation does not universally predict heightened ALT activity across various cancer types,” citing large-scale pan-cancer analyses and experimental studies that support this interpretation. We further underscore that ALT activity is context-dependent and likely requires additional contributing factors beyond ATRX loss.
Importantly, in our analysis of ALT pathway activity, we do not rely solely on the presence or absence of specific mutations (e.g., ATRX/DAXX/p53/H3.3), but instead adopt a pathway-based approach that incorporates network topology, pathway directionality, and PSF scores to capture functional activity in a systems-level context.
Additionally, we have acknowledged this ambiguity in the Discussion Limitations section [Pg 14, Ln 21-22 ], noting that “the ambiguous role of ATRX in ALT telomere maintenance and the ambiguous interpretation of data pose significant challenges.”